# *Sporothrix globosa* melanin regulates autophagy via the TLR2 signaling pathway in THP-1 macrophages

**Mengqi Guan**[1], **Lei Yao**[1], **Yu Zhen**[1], **Yang Song**[1], **Xiaobo Liu**[2], **Yuanyuan Liu**[1], **Ruili Chen**[1,3], **Yan Cui**●[1]*, **Shanshan Li**[1]*

1 Department of Dermatology and Venereology, First Hospital of Jilin University, Changchun, China,
2 Laboratory of Cancer Precision Medicine, First Hospital of Jilin University, Changchun, China,
3 Department of Dermatology and Venereology, Zhuhai People's Hospital, Zhuhai, China

* supermandsdzam@jlu.edu.cn (YC); lishans@jlu.edu.cn (SL)

**Data Availability Statement:** All relevant data are within the manuscript and its Supporting Information files.

## Abstract

Melanin, an important virulence factor of pathogenic fungi, has been shown to suppress host immune responses in multiple ways. Autophagy is a vital cellular mechanism underlying the host's innate immunity against microbial infections. However, the potential influence of melanin on autophagy has not been explored. We investigated the effect of melanin on autophagy in macrophages, which play a key role in controlling *Sporothrix* spp. infection, as well as the mechanism of melanin interaction with Toll-like receptor (TLR)-induced pathways. *Sporothrix globosa* conidia (wild-type and melanin-deficient mutant strains) or yeast cells were co-cultured with THP-1 macrophages to demonstrate that, although *S. globosa* infection led to the activation of autophagy-related proteins and increased autophagic flux, *S. globosa* melanin suppressed macrophage autophagy. Incubation with *S. globosa* conidia also increased the expression levels of reactive oxygen species and multiple proinflammatory cytokines (interleukin-6, tumor necrosis factor-α, interleukin-1β and interferon-γ) in macrophages. These effects were attenuated as melanin presented. Furthermore, while *S. globosa* conidia significantly increased the expression of both TLR2 and TLR4 in macrophages, the knockdown of TLR2, but not TLR4, with small interfering RNA suppressed autophagy. Overall, this study revealed the novel immune defense ability of *S. globosa* melanin to inhibit macrophage functionality by resisting macrophage autophagy through the regulation of TLR2 expression.

## Author summary

Melanin as an important virulence factor for *Sporothrix globosa*, analysis of the effects on host immunity, is of great significance for guidance regarding future directions of studies on the mechanism underling fungal pathogenesis. In recent years, accumulating evidence supports a role for autophagy as a host defense mechanism to counteract the evasion strategies of viral, bacterial and parasitic pathogens. Our study aimed to investigate the effect of melanin on autophagy in macrophages, with particular attention in *Sporothrix spp*. to

**Funding:** This work was supported by the National Natural Science Foundation of China (Grant No. 82073454 for SL), the Natural Science Foundation of Jilin Province (Grant Nos. 20200201485JC for SL and 20210204190YY for YC) and the Finance Department Health Special Project of Jilin Province (JLSWSRCZX2020-007 for SL). The funders had no role in study design, data collection and analysis, decision to publish, or preparation of the manuscript.

**Competing interests:** The authors have declared that no competing interests exist.

explore the mechanism of melanin interaction with Toll-like receptor (TLR)-induced pathways. In addition, this work also served as an important theoretical supplement for our previous research on *S. globosa* melanin. We believe that our research makes a significant contribution to the melanin pigment in fungi because although a lot of progress is being made in the field of pathogenic fungi, most studies are in animal macrophages in vitro studies; exploring the macrophage autophagy will provide a novel immune defense ability of *S. globosa* melanin to inhibit macrophage.

## 1. Introduction

Melanin is well established as a crucial virulence factor that resides in the cell walls of pathogenic fungi [1, 2]. To adapt to continuously changing environments, fungal melanin may resist extreme temperatures and heavy metal toxicity, defend against UV radiation, and neutralize reactive oxygen species (ROS) and other radicals [3, 4]. Additionally, in *Paracoccidioides brasiliensis*, melanin enhances antifungal resistance by binding to antifungal agents [5]. Previous studies of fungal melanin mostly focused on its capacity to protect against functions of mammalian immune systems, such as phagocytosis and killing. For example, phagocytosis and pathogen killing are reduced in melanized cells compared with non-melanized cells from organisms that include *Talaromyces marneffei*, *Fonsecaea pedrosoi* and *Cryptococcus neoformans* [6–8]. Recently, elaborate mechanisms underlying the melanin-mediated inhibition of phagocytosis have been revealed. Akoumianaki and coworkers [9] demonstrated that *Aspergillus* melanin selectively blocked nicotinamide adenine dinucleotide phosphate oxidase-dependent activation of LC3-associated phagocytosis (LAP), a noncanonical form of autophagy, suggesting that more research is needed to explore the relationship between melanin and autophagy.

Autophagy is a lysosome-mediated catabolic process within the cytosol of double-membrane vesicles in eukaryotes, whereby ingestion and digestion of damaged organelles and unused proteins is utilized to maintain cellular homeostasis, supporting cell survival and regulating inflammation [10–12]. In recent years, accumulating evidence supports a role for autophagy as a host defense mechanism to counteract the evasion strategies of viral, bacterial and parasitic pathogens [13]. Despite previous studies showing that *Candida albicans*, *C. neoformans* and *Aspergillus fumigatus* can induce autophagy (or LAP), thereby suggesting that this process participates in antifungal immunity, the role of autophagy in innate responses against fungi remains elusive [14–16]. Qin and colleagues demonstrated that autophagy induced by *C. neoformans* in macrophages could be inhibited by knocking down multiple autophagy (ATG) proteins or 3-methyladenine (3-MA), an autophagy inhibitor, contributing to reduced phagocytosis [17]. By contrast, *C. albicans*-induced autophagy did not affect phagocytosis [16, 18]. The above studies indicate that the role of autophagy in manipulating immune responses varies greatly across different species of fungi [13]. A similar outcome was also found by Nicola and colleagues, that myeloid cells with ATG5 deficiency exhibited different susceptibility to *C. albicans* and *C. neoformans* infections [14]. To date, little research has analyzed the role of autophagy in the pathogenesis of other pathogenic fungi.

Sporotrichosis is a subacute or chronic infection that mainly involves the skin and subcutaneous tissue with neighboring lymphatics, and is caused by dimorphic fungi of pathogenic *Sporothrix* species, including *S. schenckii*, *S. brasiliensis* and *S. globosa* [19–21]. The pathogenesis of sporotrichosis remains largely obscure [22]. Melanin, an important virulence factor of *Sporothrix* spp., is synthesized as three main types by both conidia and yeast cells:

1,8-dihydroxynaphthalene (DHN)-melanin, eumelanin and pyomelanin [23, 24]. Studies showed that these melanins could inhibit the phagocytosis of macrophages and reduce the susceptibilities of melanized cells to antifungal agents [23, 25, 26]. Interestingly, our previous work revealed that *S. globosa* melanin could shape adaptive immunity by decreasing the expression of major histocompatibility complex class II and costimulatory molecules, leading to impairment of antigen-presenting capability in murine macrophages [27]. Moreover, we recently discovered that the Toll-like receptor (TLR)2 and TLR4 signaling pathways are involved in regulating the protective effect of *S. globosa* melanin against phagocytosis by THP-1 cells [28].

In the present study, we used melanin-producing (Mel+) strains, melanin-deficient (Mel-) mutant strains and yeast cells of *S. globosa* to evaluate the impact of melanin on autophagy of *S. globosa* by macrophages. We employed THP-1 macrophages to demonstrate that autophagy can be induced by *S. globosa* and to investigate the possible signaling pathway.

## 2. Materials and methods

### 2.1. Cell culture

Human monocytic THP-1 cells purchased from the American Type Culture Collection (Manassas, VA, USA; LOT: 70013348) were cultured in RPMI 1640 medium supplemented with 10% fetal bovine serum and 1% penicillin–streptomycin at 37˚C in a humidified incubator containing 5% $CO_2$. All cell culture reagents were obtained from Gibco Laboratories (Gibco, Grand Island, NY, USA). THP-1 macrophages were prepared by stimulating the monocytic cells with 100 ng/mL phorbol 12-myristate 13-acetate (PMA; Sigma, St. Louis, MO, USA) for 24 h before use. For infection experiments, THP-1 macrophages were seeded in 6-well plates at a density of $1 \times 10^6$ cells/well to grow adherently overnight.

### 2.2. Fungal strains and suspension preparation

The wild-type strain of *S. globosa* (Mel+, FHJU12082703) and a melanin-deficient mutant (Mel-), which were identified based on phenotypic characteristics and the nucleotide sequence of the calmodulin (CAL) gene (GenBank accession no. KT008664), were stored in our laboratory and used for all experiments [27–29]. The Mel+ strain was obtained from a patient with cutaneous-lymphatic sporotrichosis, and the Mel- strain was obtained by UV mutagenesis. The strains were recovered on potato dextrose agar slants (BD Difco, Sparks, MD, USA) at 28˚C for 7 days before use. To obtain yeast cells of *S. globosa*, the Mel+ strain was consecutively subcultured twice on brain–heart infusion agar (BHI, BD Difco) at 37˚C for 7 days. *S. globosa* conidia and a yeast cell suspension were collected in 0.05% PBST (0.01 M phosphate-buffered saline containing 0.05% Tween 80), filtered through eight gauze layers, and enumerated using a hemocytometer.

### 2.3. THP-1 macrophage infection

THP-1 macrophages were prepared as described above and stimulated with *S. globosa* (Mel + or Mel-) conidia or yeast cells for the indicated time points at different concentrations. Where appropriate, cells were pre-incubated with 20 μM chloroquine, 500 nM wortmannin, or 5 μM rapamycin (all from MedChemExpress, Shanghai, China) for different time periods. Contamination of LPS for all agents was excluded by *Limulus* Amebocyte Lysate Assay Kit (Xiamen Bioendo Technology, Xiamen, China).

## 2.4. Western blot analysis

Total protein was collected from cells using RIPA Lysis Buffer (Beyotime, Suzhou, China). Whole protein extracts were separated by 12% sodium dodecyl sulfate-polyacrylamide gel electrophoresis and transferred onto polyvinylidene difluoride membranes (Millipore Corporation, Temecula, CA, USA). Western blotting was performed according to the manufacturer's instructions using the following primary antibodies: rabbit anti-LC3A/B at 1:1000 (Cat. #12741, Cell Signaling Technology, Beverly, MA, USA), rabbit anti-Beclin1 at 1:1000 (Cat. #3495, Cell Signaling Technology), rabbit anti-SQSTM1/p62 at 1:1000 (Cat. #8025, Cell Signaling Technology), mouse anti-GAPDH antibody at 1:4000 (Cat. # T0004, Affinity Biosciences, Cincinnati, OH, USA), rabbit anti-TLR2 antibody at 1:1000 (Cat. # 12276, Cell Signaling Technology), rabbit anti-TLR4 antibody (Cat. # BS3489, Bioworld Technology, Minneapolis, MN, USA) at 1:500 and rabbit anti-Atg7 antibody at 1:1000 (Cat. # DF6130, Affinity Biosciences). After washing three times in TBST solution, the membranes were incubated with the respective secondary antibodies (anti-mouse HRP or anti-rabbit HRP at 1:4000, Affinity Biosciences) for 1 h at room temperature. The blots were assessed by enhanced chemiluminescence detection (Affinity Biosciences). The mean pixel density of each protein band was quantified using Image J (NIH Image J system, 1.8.0). Anti-GAPDH expression was analyzed as an internal control, and protein expression was quantified as the ratio of the specific band to GAPDH.

## 2.5. Analysis of fungal killing assays

The fungal killing assay was conducted according to a previously published protocol [28]. Briefly, THP-1 macrophages ($2 \times 10^4$ cells/well in a 24-well plate), in the presence or absence of chloroquine (20 μM) or rapamycin (5 μM), were incubated with *S. globosa* (Mel+ or Mel-) conidia (multiplicity of infection [MOI] = 10) in complete medium. After 24 h of infection, the conidia outside of the cells were killed by nystatin (Toronto Research Chemicals, Toronto, ON, Canada) at 50 mg/mL. Conidia incubated with cells at 0 h were introduced as a negative control. Then, cells were lysed with 0.5% Triton X-100 and diluted. The number of viable *S. globosa* conidia was determined by enumeration of the colony-forming units (CFUs) on Sabouraud dextrose agar plates after incubation at 28°C for 7 days. All samples were tested in triplicate and strain counts were expressed as the Log CFU/mL.

## 2.6. Transfection and confocal microscopy of mRFP-GFP-LC3 adenovirus

THP-1 cells were transfected with an adenovirus construct carrying LC3 tagged with monomeric red fluorescent protein (mRFP) and green fluorescent protein (GFP) (mRFP-GFP-LC3; HanBio, Wuhan, China), which expresses a specific marker of autophagosome formation used to detect autophagic flux, as previously described [30, 31]. Briefly, THP-1 cells cultured on cover slips were incubated with adenovirus particles for 8 h post-infection; the culture medium was replaced and the incubation continued for 36 h; and cells were transfected with control small interfering (si)RNA, TLR2 siRNA, TLR4 siRNA, or no treatment as a control. The THP-1 cells were then challenged with *S. globosa* (Mel+ or Mel-) conidia (MOI = 10). Laser scanning confocal microscopy (Olympus FV3000, Tokyo, Japan) was employed to visualize and record images of the autophagic flux. All image acquisition settings were kept the same during image collection. The yellow and red spots observed after image overlapping represented autophagosomes and autolysosomes, respectively. The numbers of spots of different colors were determined by manual counting of fluorescent puncta from at least 50 cells, and experiments were repeated independently at least four times.

## 2.7. Determination of ROS

A dichlorofluorescein assay was used to measure ROS [32]. The principal component of this assay is 2,7-dichlorodihydrofluorescein diacetate (DCFH-DA), which is easily oxidized to fluorescent dichlorofluorescein (DCF) by intracellular ROS, thereby enabling the quantification of ROS levels. Briefly, THP-1 macrophages ($2 \times 10^5$/well) were plated into 96-well black round-bottom plates (Corning, New York, NY, USA) and infected with *S. globosa* (Mel+ or Mel-) conidia (MOI = 10). After a 12-h incubation, 2 DCFH-DA (Beyotime) was added to each well to a final concentration of 10 μmol/L, and the cells were incubated at 37˚C in a dark room for 20 min. The results were obtained using a fluorescence plate reader at 488 nm for excitation and 535 nm for emission. The test was performed in triplicate.

## 2.8. Cytokine release assay

THP-1 macrophages were cultured in the presence or absence of chloroquine (20 μM) or rapamycin (5 μM) and then the cells were infected with Mel+ or Mel- *S. globosa* conidia. After 12 h, supernatants were collected and interleukin (IL)-6, IL-1β, interferon (IFN)-γ and tumor necrosis factor (TNF)-α were measured using commercial ELISA kits (Thermo Fisher Scientific, Waltham, MA, USA), according to the manufacturer's instructions. Standard curves from each test were used to calculate cytokine concentrations.

## 2.9. Transfection of siRNA

THP-1 macrophages were transiently transfected with TLR2-siRNA, TLR4-siRNA, negative control (ctrl)-siRNA (Cat. # SI00050022; Cat. # SI04951135, Cat. # 1027280, respectively; Qiagen, Germantown, MD, USA) and Atg7-siRNA (Cat. #RX066645, Public Protein/Plasmid Library, Nanjing, China). Adherent cells were transfected with siRNA using Hiperfect transfection reagent (Qiagen) according to the manufacturer's instructions. Knockdown efficiency was determined by western blot analysis. Three independent transfection experiments were performed.

## 2.10. Statistical analysis

All data are expressed as the mean ± standard deviation (SD). An unpaired t-test was used to analyze the differences between two groups; Parametric (Bonferroni's post-test or Student-Newman-Keuls test (S-N-K test)) and non-parametric (Kruskal–Wallis test) ANOVA were used as appropriate to compare three or more groups. All statistical analyses were conducted using SPSS 20.0 (IBM Corp., Armonk, NY, USA) and a significance level of 5% ($P < 0.05$) was adopted. Graphic data interpretation was performed using GraphPad Prism 7.0 (GraphPad Software Inc., San Diego, CA, USA).

# 3. Results

## 3.1. *S. globosa* infection induced autophagy in THP-1 macrophages

To determine whether infection with *S. globosa* could induce autophagy, we examined changes in autophagosomal proteins and autophagic flux in THP-1 macrophages. First, we used western blotting to assay the levels of autophagy marker proteins LC3-II/I, Beclin1 and p62 after exposure to *S. globosa* (Mel+ or Mel-) conidia, yeast cells (of a no melanin-producing strain) or rapamycin (5 μM) at MOIs ranging from 1 to 100, at five time points between 2 and 24 h. Obvious enhancements in the expression of Beclin1 and LC3-II proteins, and a decrease in the level of protein p62 occurred in THP-1 macrophages, showing dynamic changes over time and across the range of concentrations. The expression level of LC3 reached its peak at an MOI of 100, while those of Beclin1 and p62 peaked at an MOI of 10 for *S. globosa* conidia, but

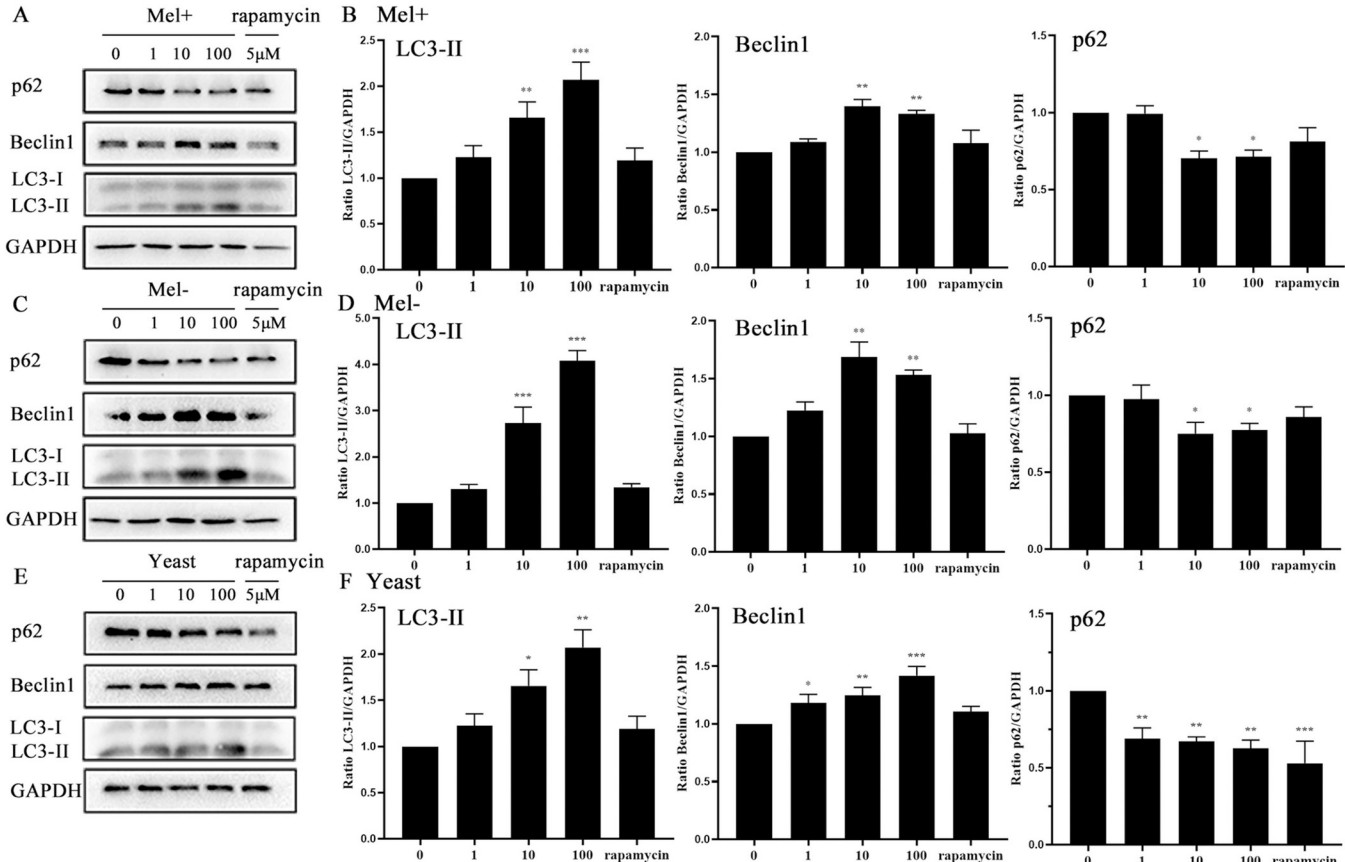

**Fig 1. Protein levels in macrophages infected with *S. globosa* (Mel+ or Mel-) conidia or yeast cells at different MOI.** THP-1 macrophages pretreated with rapamycin (5 μM, 12 h) and subsequently infected at different MOIs of Mel+ (A), Mel- (C) *S. globosa* conidia, or yeast cells (E) for 12 h were used for western blotting of LC3II/LC3I, Beclin1 and p62. (B, D, and F) The calculated ratios of mean grayscale scanning of LC3II, Beclin1 and p62 to GAPDH are shown; *$P < 0.05$, **$P < 0.01$, ***$P < 0.001$, for one-way ANOVA followed by S-N-K test. Data represent three experiments with similar results and representative images are shown.

for yeast cells the protein levels all peaked at an MOI of 100 (Fig 1A–1F). In general, the protein expression levels of LC3-II and Beclin1 increased over time, while the p62 level decreased in *S. globosa* strains infected with THP-1 cells (Fig 2A–2F).

Next, we utilized the tandem fluorescent-tagged RFP-GFP-LC3 adenovirus construct to monitor autophagic flux by analyzing the extent of autophagosome and autolysosome formation, as described previously [30, 31]. This assay capitalizes on the pH difference between acidic autolysosomes and neutral autophagosomes, and the pH-sensitivity differences exhibited by GFP and RFP to monitor progression from the autophagosome to the autolysosome. When an autophagosome fuses with a lysosome to form autolysosomes, the GFP moiety degrades. Briefly, the red dots that overlay green dots appear yellow in merged images and are indicators of autophagosomes, whereas the red dots that do not overlay green dots and appear red in merged images indicate autolysosome formation. Fig 3 shows that RFP-GFP-LC3 was successfully introduced into THP-1 macrophages, which produced both fluorescent proteins. In addition to the accumulation of LC3, more red puncta were present in THP-1 macrophages infected with Mel- conidia (MOI = 10) than in those infected with Mel+ conidia or uninfected cells. These results suggested both Mel+ and Mel- *S. globosa* could induce autophagy in THP-1 macrophages.

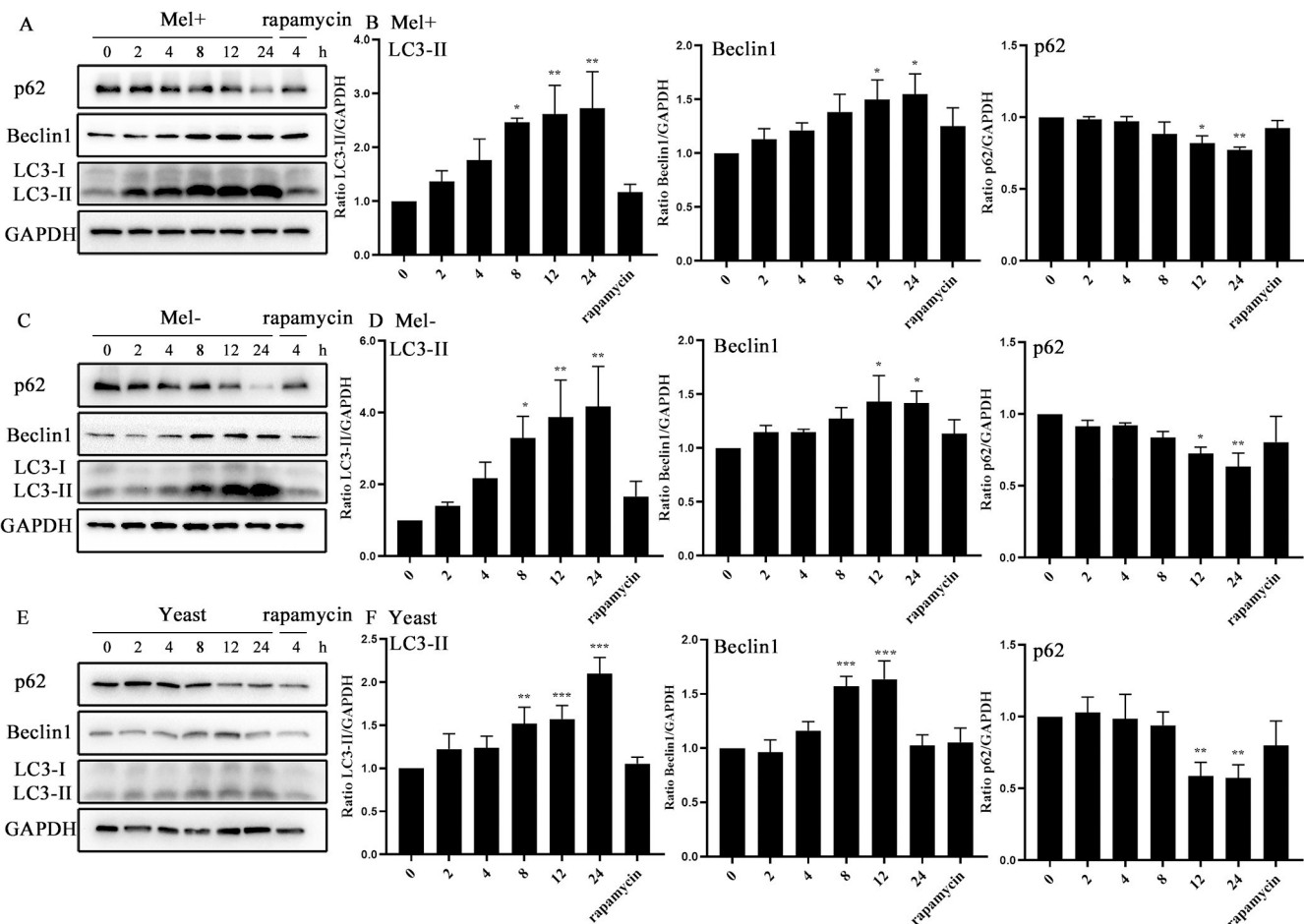

**Fig 2. Protein levels in macrophages infected with *S. globosa* (Mel+ or Mel-) conidia or yeast cells for different time.** THP-1 macrophages pretreated with 5 μM rapamycin for 12 h and then incubated with Mel+ (A), Mel- (C) *S. globosa* conidia, or yeast cells (E) (MOI = 10) for different indicated times were used for western blotting of LC3II/LC3I, Beclin1 and p62. (B, D, and F) The calculated ratios of mean grayscale scanning of LC3II, Beclin1 and p62 to GAPDH are shown; *P < 0.05, **P < 0.01, ***P < 0.001, for one-way ANOVA followed by S-N-K test. Data represent three experiments with similar results and representative images are shown.

Finally, to determine whether the induction of autophagy could kill *S. globosa*, we performed killing assay. The results showed that the number of CFUs of *S. globosa* (Mel+ or Mel-) decreased after 12 h of co-incubation with macrophages. Moreover, Mel- cells showed lower survival rates than Mel+ cells, and statistically significant interactions were found between the chloroquine and rapamycin groups regardless of the presence of melanin. This indicated that autophagy may promote the clearance of *S. globosa* in THP-1 macrophages (Fig 4).

### 3.2. The presence of melanin inhibited autophagy induced by *S. globosa* in THP-1 macrophages

To explore the effect of Mel+ versus Mel- *S. globosa* conidia or yeast cells on the induction of autophagy in THP-1 macrophages, we examined the expression of autophagy-related markers by western blotting. Fig 5 revealed that expression of LC3-II in THP-1 macrophages was enhanced in the absence of melanin, with the Mel- conidia and yeast cell group showing remarkably higher expression levels than the Mel+ conidia group (*P* < 0.05). The expression level of Beclin1 protein in the Mel- conidia group was higher than that in the Mel+ conidia

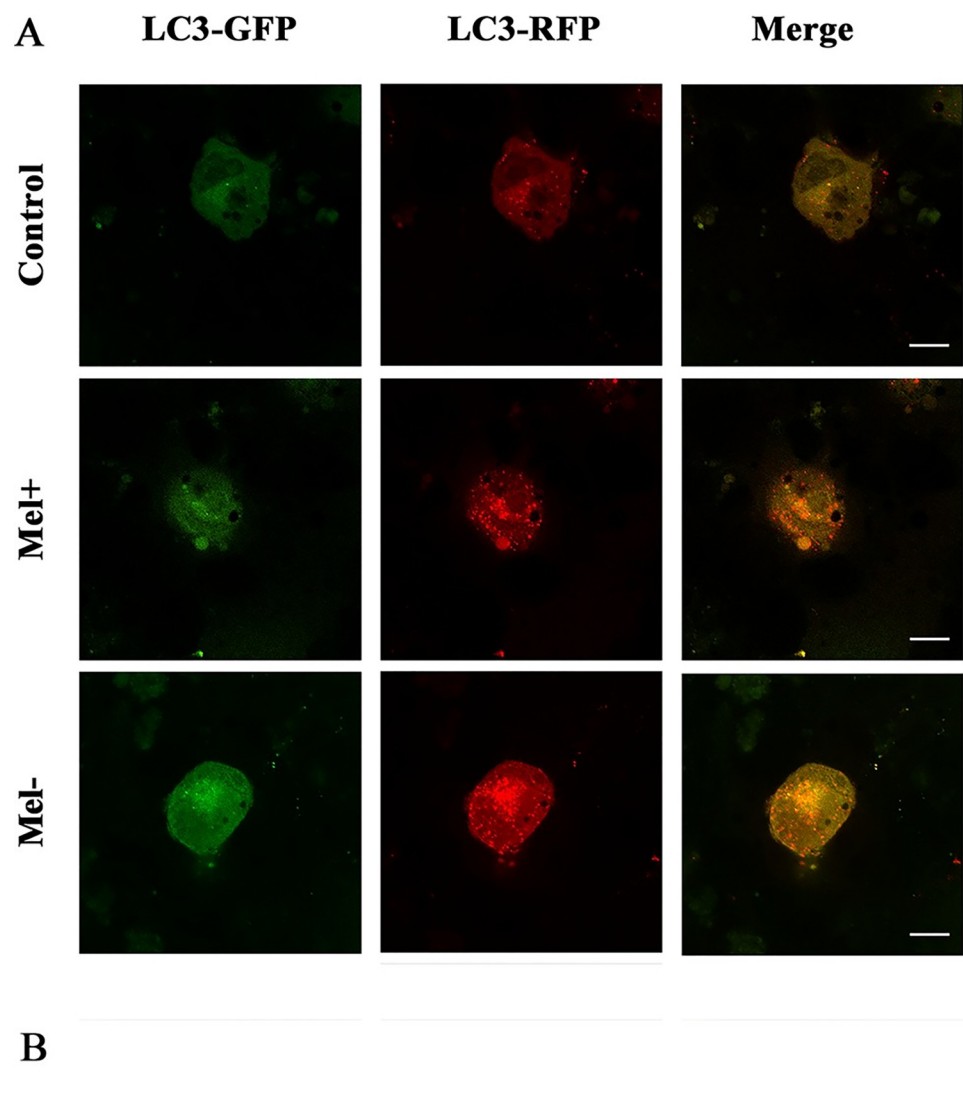

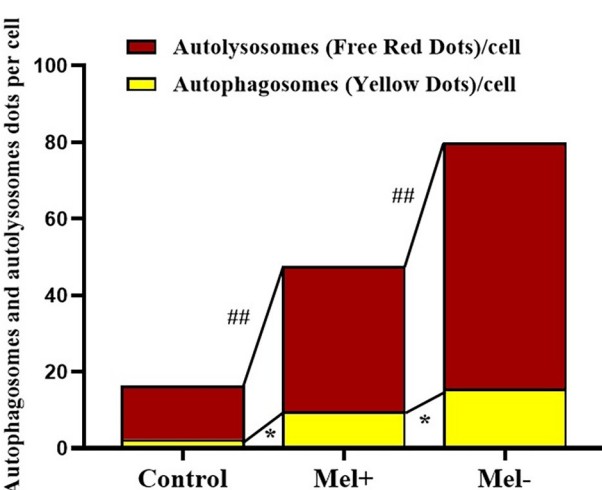

**Fig 3. Fluorescent LC3 puncta following transfection with mRFP-GFP-LC3 adenovirus in macrophages infected with *S. globosa* conidia.** (A) Confocal microscopy of mRFP-GFP-LC3 adenovirus-infected THP-1 macrophages subsequently infected with Mel+ or Mel- *S. globosa* conidia (MOI = 10) for 12 h. Scale bar = 10 μm. (B) Calculated

numbers of autolysosomes (red) and autophagosomes (yellow). Data represent the mean from four independent experiments; $^{\#\#}P < 0.01$ indicates between-group differences in autolysosomes (red); $^{*}P < 0.05$ indicates between-group differences in autophagosomes (yellow). One-way ANOVA followed by Bonferroni's test. The numbers of spots of different colors were determined by manual counting of fluorescent puncta from at least 50 cells, and experiments were repeated independently at least four times. Finally, representative images were selected.

group. By contrast, no obvious differences were detected in the levels of p62 and Beclin1 between Mel+ conidia and yeast cells. Furthermore, autophagic flux detected by confocal microscopy showed that the numbers of both autophagosomes (yellow) and autolysosomes (red) were significantly higher in THP-1 macrophages infected with Mel- conidia compared with those in Mel+ conidia (Fig 3). Meanwhile, as shown in Fig 4, the killing effect was strongly

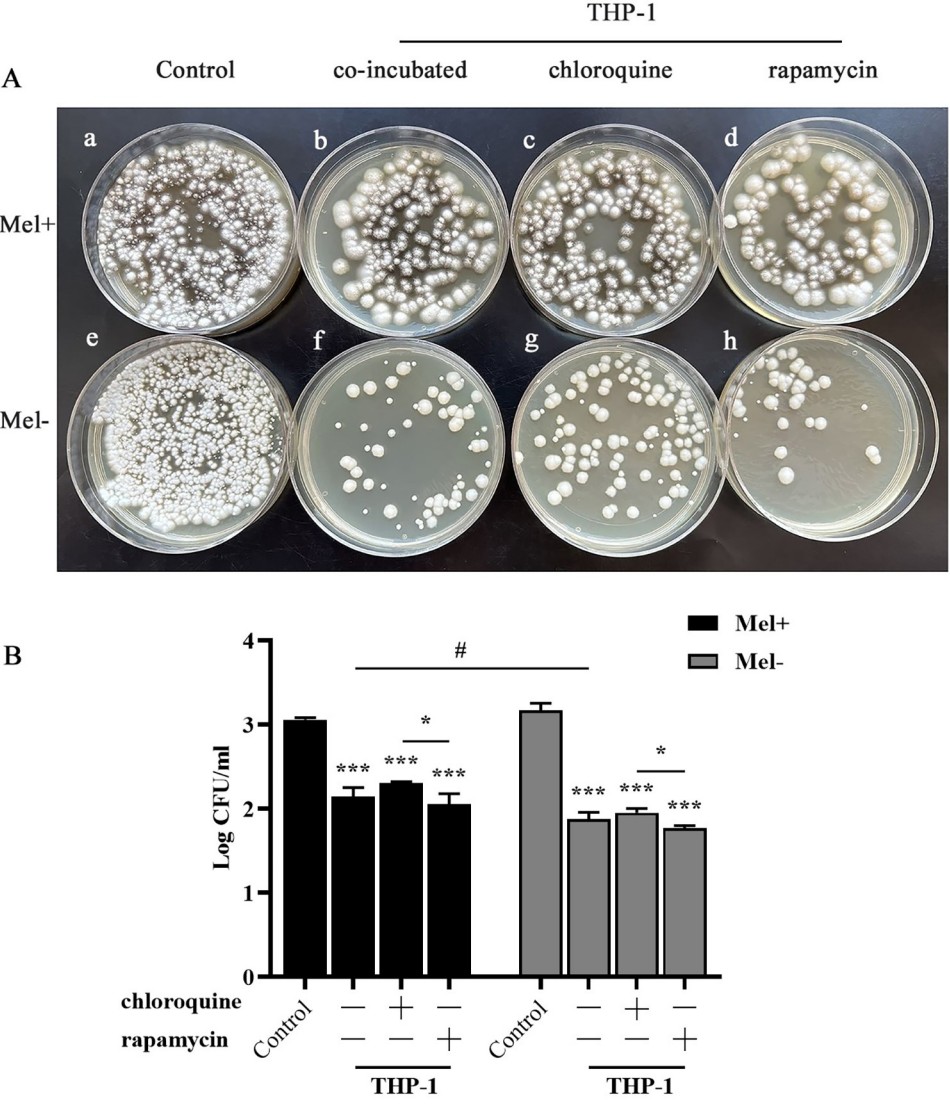

**Fig 4. Killing assay of *S. globosa* conidia in activated THP-1 macrophage autophagy.** (A) Colonies of the Mel+ (a–d) and Mel- (e–h) strains pretreated with or without chloroquine (20 μM, 4 h) or rapamycin (5 μM, 12 h) and THP-1 macrophages for 12 h, and then subcultured on Sabouraud dextrose agar at 28˚C for 7 days. (B) Strain counts were expressed as the Log CFU/mL. Bars represent the mean ± SD of three independent samples. $^{*}P < 0.05$ and $^{***}P < 0.001$, for one-way ANOVA followed by Bonferroni's post-test; $^{\#}P < 0.05$, for unpaired t-test.

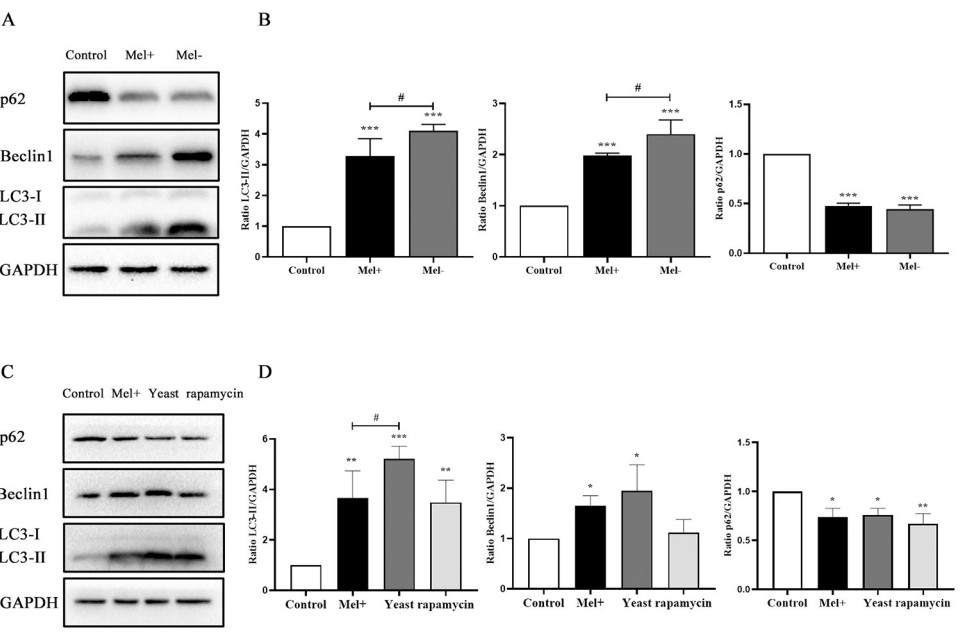

**Fig 5. Comparison of *S. globosa* (Mel+ or Mel-) conidia and yeast cells in the induction of autophagy in macrophages.** (A, C) Western blotting of LC3-II, Beclin1 and p62 proteins in THP-1 macrophages pretreated with or without rapamycin (5 μM, 12 h) and infected with Mel+ conidia, Mel- conidia or yeast cells (MOI = 10) for 12 h. (B, D) The ratios of LC3-II/GAPDH, Beclin1/GAPDH and p62/GAPDH were calculated; $^*P < 0.005$, $^{**}P < 0.01$, $^{***}P < 0.001$, $^\#P < 0.05$, for one-way ANOVA followed by Bonferroni's test. Representative images are shown.

decreased in the presence of melanin, with higher numbers of CFUs in the Mel+ conidia group compared with the Mel- conidia group.

### 3.3. Effects of chloroquine, wortmannin and rapamycin on autophagy induced by *S. globosa* in THP-1 macrophages

We examined the impacts of the following modulators on autophagy induced by *S. globosa*: wortmannin (500 nM), an early-phase inhibitor of autophagy flux; chloroquine (20 μM), a late-phase inhibitor of the autophagy process; and rapamycin (5 μM), a mammalian target of rapamycin (mTOR) inhibitor and autophagy activator. As shown in Fig 6A, THP-1 cells pretreated with chloroquine or wortmannin both showed increased expression of p62 protein and reduced expression of Beclin1 protein. Our results also showed that LC3-II accumulated under treatment with chloroquine and rapamycin and LC3-I mainly increased under wortmannin treatment in Fig 6B. Rapamycin treatment increased the protein level of Beclin1 and decreased the protein level of p62 in Fig 6C-6D. Of note, *S. globosa*-infected cells pretreated with rapamycin showed reduced expression of LC3-II, which might be due to the potent induction of autophagy resulting in an increase in LC3-II degradation [33].

### 3.4. *S. globosa* stimulated ROS production and proinflammatory cytokine secretion in THP-1 macrophages undergoing autophagy

Next, we investigated the possible effects of autophagy induced by *S. globosa*, which include the generation of ROS. As shown in Fig 7A, infection of THP-1 macrophages with Mel+ or Mel- *S. globosa* conidia triggered a large increase in ROS production compared with that in the uninfected control group. In cells infected by both Mel+ and Mel- *S. globosa* conidia, pretreatment with chloroquine (20 μM) inhibited ROS production, whereas pretreatment with

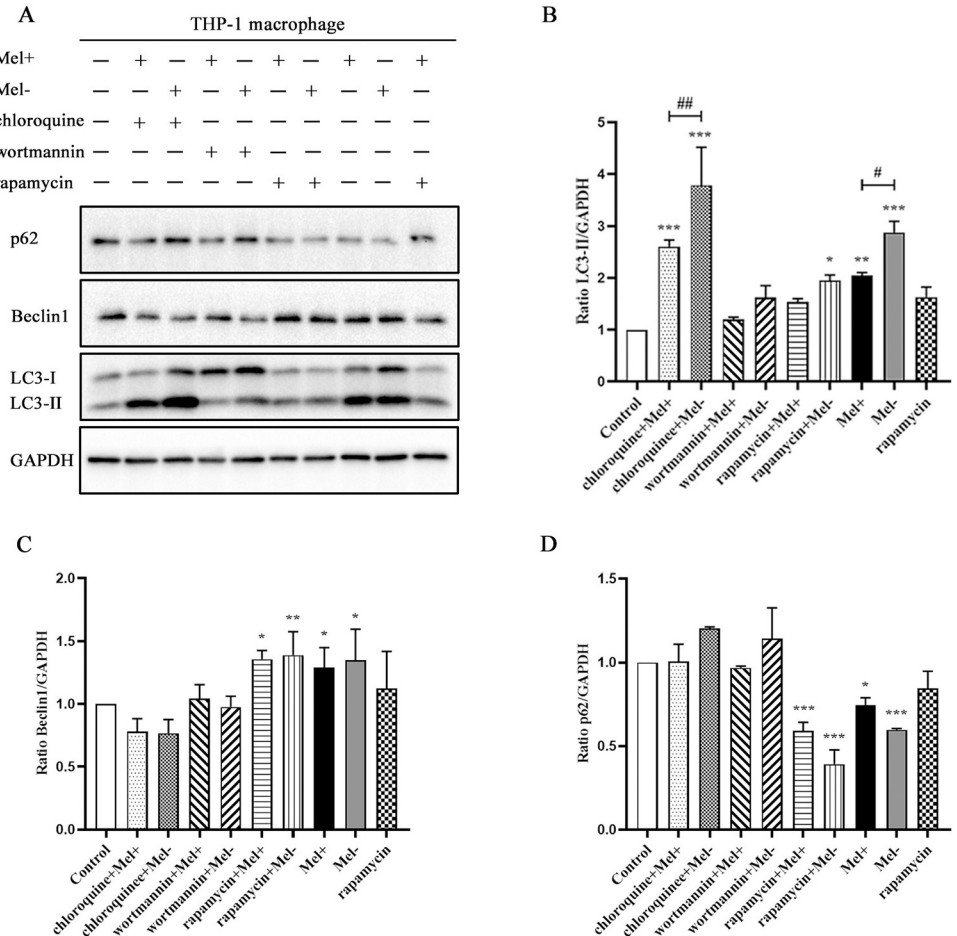

**Fig 6. Effects of chloroquine, wortmannin and rapamycin treatment on autophagy induced by *S. globosa* in macrophages.** (A) Western blotting of THP-1 macrophages pretreated with or without chloroquine (20 μM, 4 h), wortmannin (500 nM, 4 h) or rapamycin (5 μM, 12 h), and subsequently infected with Mel+ or Mel- *S. globosa* conidia (MOI = 10) for 12 h. (B–D) The ratios of LC3-II/GAPDH, Beclin1/GAPDH and p62/GAPDH were calculated; *$P < 0.005$, **$P < 0.01$, ***$P < 0.001$; #$P < 0.05$, ##$P < 0.01$, for non-parametric one-way ANOVA followed by the Kruskal–Wallis test. Representative images are shown.

rapamycin (5 μM) led to a trend toward even higher production of ROS. Additionally, ROS production was enhanced in the absence of melanin, with the Mel- conidia group showing remarkably higher production compared with the Mel+ conidia group.

We also evaluated the levels of four proinflammatory factors (TNF-α, IL-6, IL-1β and IFN-γ) in THP-1 macrophages preincubated with chloroquine (20 μM) or rapamycin (5 μM), followed by infection with Mel+ or Mel- *S. globosa* conidia to activate autophagy. At 12 h post-infection, THP-1 macrophages secreted higher levels of TNF-α, IL-6, IL-1β and IFN-γ compared with untreated/uninfected controls (Fig 7B–7E), with the Mel- conidia group showing significantly higher levels of all four factors compared with the Mel+ conidia group ($P < 0.05$). By contrast, cells that were pretreated with chloroquine before infection exhibited trends towards decreasing levels of proinflammatory TNF-α, IL-6 and IFN-γ, and increasing levels of IL-1β, compared with untreated cells infected with Mel+ or Mel- *S. globosa* conidia. The results for these proinflammatory factors in cells pretreated with rapamycin before infection were completely opposite. To further corroborate the *S. globosa*-induced autophagy effects on cytokine secretion, we used a specific Atg7-siRNA to knock down the expression of Atg7 (an

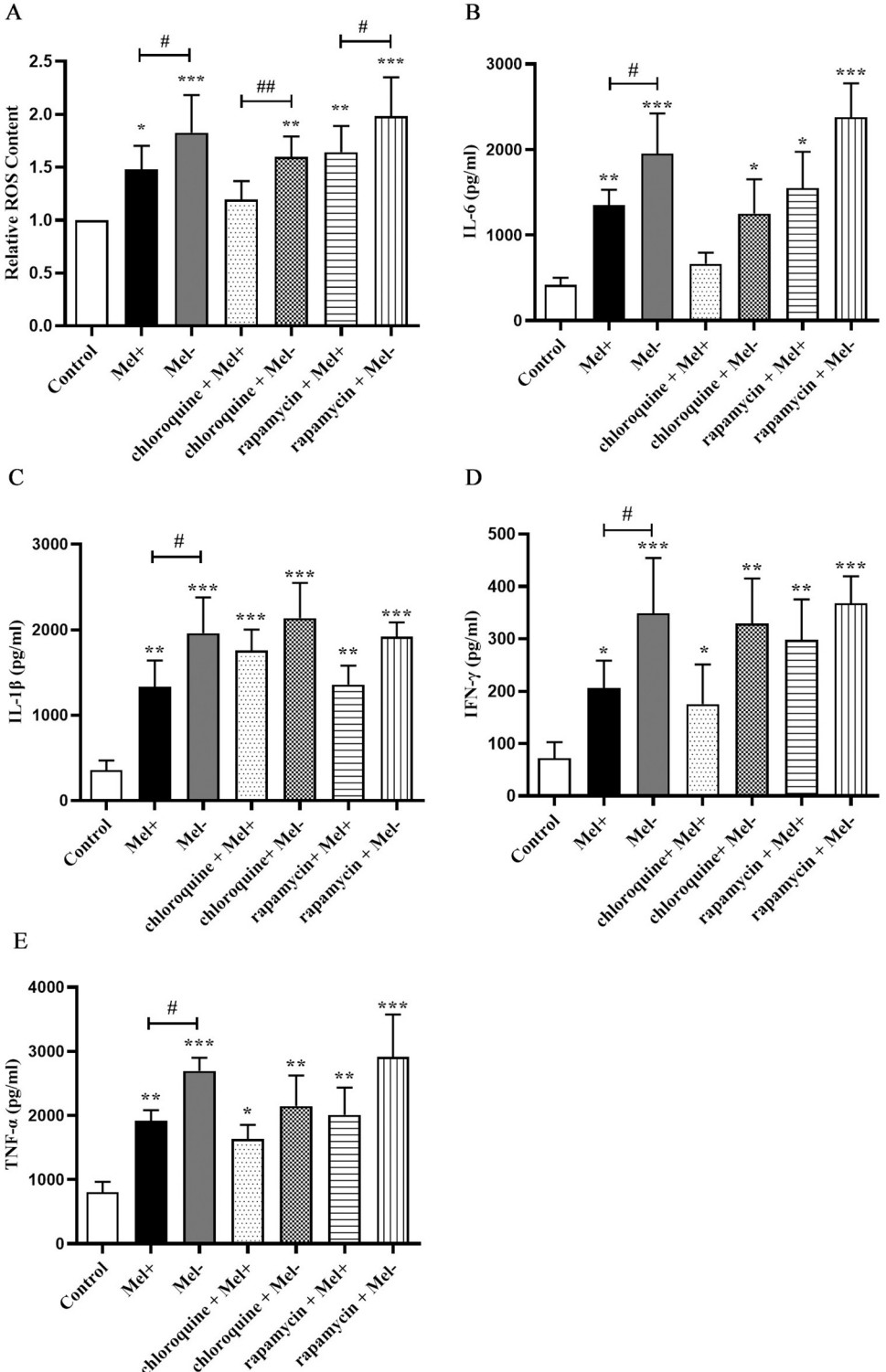

**Fig 7. Production of ROS and proinflammatory cytokines by macrophages infected with *S. globosa*.** (A). Production of reactive oxygen species (ROS) by THP-1 macrophages pretreated with or without chloroquine (20 μM, 4 h) or rapamycin (5 μM, 12 h) and subsequently infected with Mel+ or Mel- *S. globosa* conidia (MOI = 10) for 12 h. Untreated, uninfected cells served as a negative control. Data are shown as percentages relative to control cells, and as single datapoints (mean ± SD; n = 3); *$P < 0.05$, **$P < 0.01$, ***$P < 0.001$; #$P < 0.05$. (B–E) Release of inflammatory cytokines IL-6, IL-1β, IFN-γ and TNF-α in THP-1 macrophages pretreated with or without chloroquine (20 μM, 4 h)

or rapamycin (5 μM, 12 h) and subsequently infected with *S. globosa* (Mel+ or Mel-) conidia (MOI = 10) for 12 h. Untreated, uninfected cells served as a negative control. Data pooled from three sets of experiments are expressed as the mean ± SD; *$P < 0.05$, **$P < 0.01$, ***$P < 0.001$; #$P < 0.05$, for one-way ANOVA followed by Bonferroni's test.

autophagy-related gene) (S1A and S1B Fig). The expression levels of proinflammatory factors (TNF-α, IL-6 and IFN-γ) in THP-1 macrophages transfected with Atg7-siRNA were significantly decreased compared with cells transfected with Ctrl-siRNA ($P < 0.05$) (Fig 8A, 8C and 8D), whereas the IL-1β expression level showed an upward trend in Fig 8B. This result confirmed that the *S. globosa*-induced autophagic response was interfering with the expression of autophagy-specific genes downstream.

### 3.5. TLR2, but not TLR4, mediated the activation of autophagy in *S. globosa*-infected THP-1 macrophages

The protein levels of both TLR2 and TLR4 in THP-1 macrophages increased after co-culture with Mel+ or Mel- *S. globosa* conidia (S2 Fig). We next investigated whether TLR2 and TLR4 mediated *S. globosa*-infected autophagy in THP-1 macrophages that were pre-transfected with Ctrl-siRNA, TLR2 siRNA or TLR4 siRNA. As shown in S3 Fig, the expression levels of TLR2 and TLR4 were significantly lower in THP-1 macrophages pre-transfected with TLR2-siRNA or TLR4-siRNA, respectively, compared with those in cells pre-transfected with Ctrl-siRNA (Fig 9A and 9B). Furthermore, *S. globosa*-stimulated expression of LC3-II and Beclin1 was markedly attenuated, and p62 protein expression was elevated, in TLR2-siRNA cells

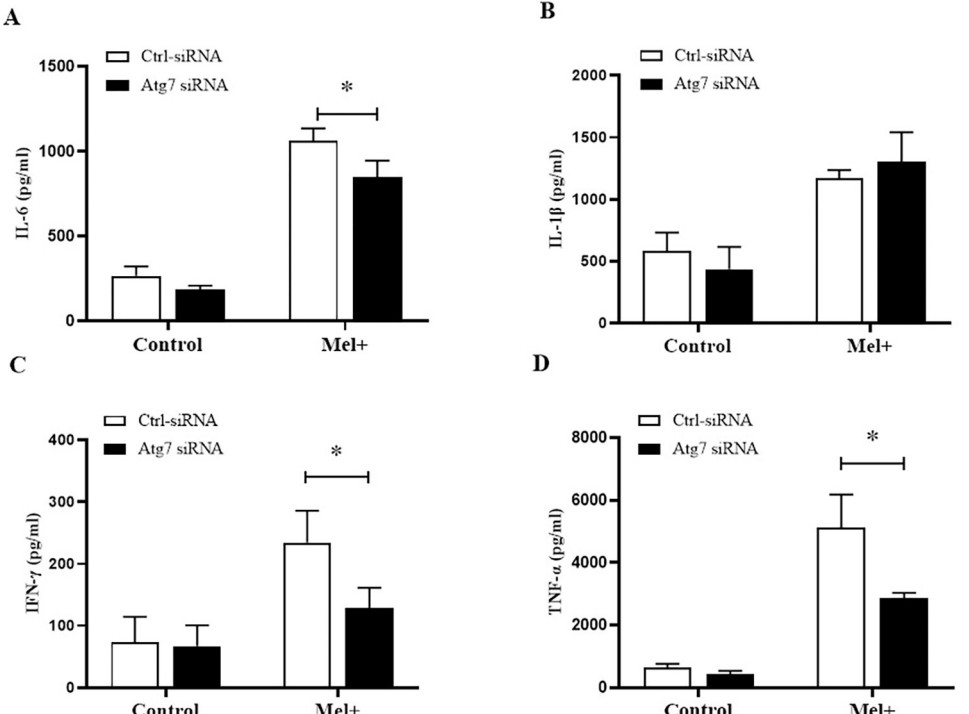

**Fig 8. Proinflammatory cytokines released by macrophages transfected with Atg7-siRNA following exposure to *S. globosa*.** THP-1 macrophages pretreated with Ctrl-siRNA or Atg7-siRNA and subsequently infected with *S. globosa* (Mel+) conidia (MOI = 10) for 12 h. Concentrations of inflammatory cytokines IL-6 (A), IL-1β (B), IFN-γ (C) and TNF-α (D) in the THP-1 macrophage groups. Data pooled from three sets of experiments are expressed as the mean ± SD; *$P < 0.05$, for the unpaired t-test.

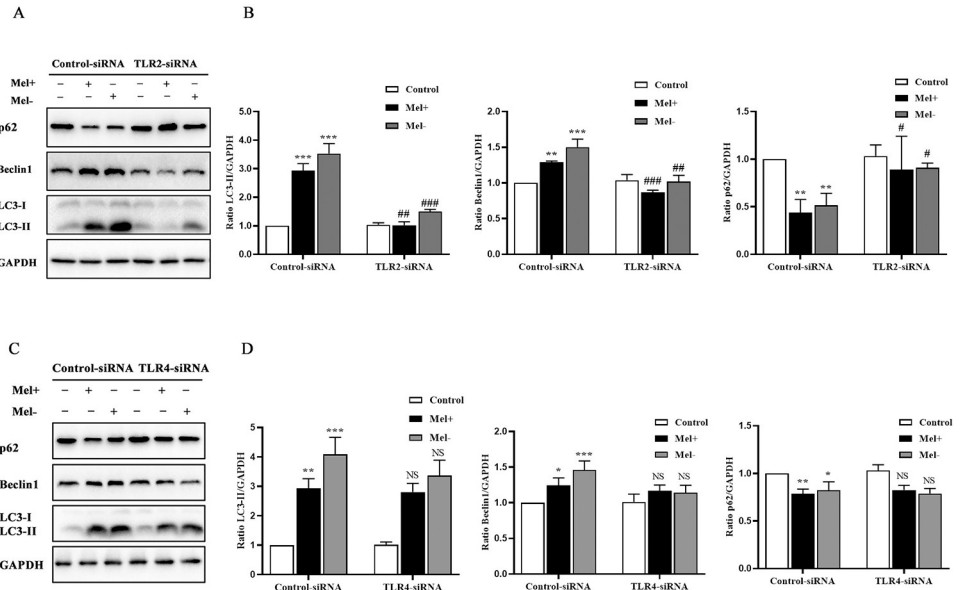

**Fig 9. Effect of TLR2 and TLR4 knockdown on autophagy in *S. globosa*-infected THP-1 macrophages.** Western blotting of autophagy markers in THP-1 macrophages pretreated with Ctrl-siRNA, TLR2-siRNA (A) or TLR4-siRNA (C) and subsequently infected with *S. globosa* (Mel+ or Mel-) conidia (MOI = 10) for 12 h. (B, D) Calculated ratios of LC3II/GAPDH, Beclin1/GAPDH and p62/GAPDH; NS represented no significant difference; *$P < 0.05$, **$P < 0.01$ and ***$P < 0.001$, for one-way ANOVA followed by Bonferroni's test; #$P < 0.05$, ##$P < 0.01$ and ###$P < 0.001$, for the unpaired t-test. Data are representative of three experiments with similar results and representative images are shown.

subsequently infected with Mel+ conidia. By contrast, there were no significant differences in protein expression of autophagy markers between TLR4-siRNA and Ctrl-siRNA cells infected with Mel+ conidia (Fig 9C, 9D). To further verify the above findings, confocal microscopy was performed to assess the numbers of autophagosomes to autolysosomes in *S. globosa*-infected THP-1 macrophages. Unsurprisingly, there were significantly fewer red and yellow puncta in Mel+ *S. globosa*-infected cells with TLR2-siRNA knockdown compared with those with Ctrl-siRNA knockdown (Fig 10A and 10C), whereas no significant differences in the numbers of autophagosomes to autolysosomes were observed in cells with TLR4-siRNA knockdown that were infected with Mel+ conidia (Fig 10B and 10D). These results demonstrated that TLR2, and not TLR4, mediated the activation of autophagy in *S. globosa*-infected THP-1 macrophages.

## 3.6. Effects of TLR2 on ROS production and proinflammatory cytokine secretion during autophagy in THP-1 macrophages infected with *S. globosa*

To demonstrate the role of TLR2 in *S. globosa*-induced autophagy, ROS production was measured in THP-1 macrophages infected with *S. globosa* after transfection with Ctrl-siRNA or TLR2-siRNA. The results demonstrated that the effect of *S. globosa*-induced autophagy on ROS production was significantly decreased following knockdown of TLR2. Correspondingly, ROS production also showed a downward trend in cells pretreated with chloroquine (20 μM) or rapamycin (5 μM) following the knockdown of TLR2 in THP-1 macrophage infection (Fig 11A).

Autophagy induction by *S. globosa* in THP-1 macrophages increased production of the proinflammatory cytokines TNF-α, IL-6, IL-1β and IFN-γ. To further investigate whether these changes would be affected by TLR2, we measured the levels of these four cytokines after

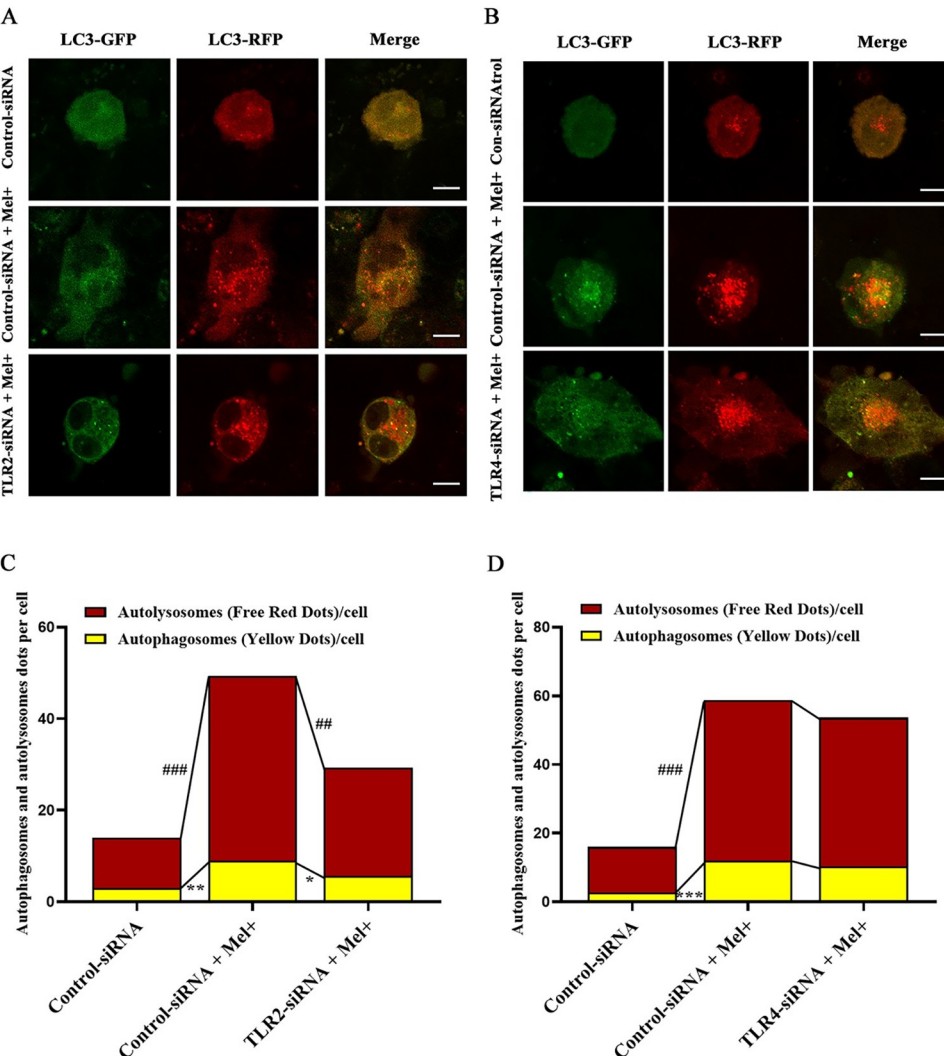

**Fig 10. Autophagy after TLR2 and TLR4 knockdown in *S. globosa*-infected macrophages.** (A, B) Confocal microscopy of THP-1 macrophages transfected with the mRFP-GFP-LC3 adenovirus for 48 h, then transfected with Ctrl-siRNA, TLR2-siRNA or TLR4-siRNA for 48 h, followed by infection with Mel+ *S. globosa* conidia (MOI = 10) for 12 h. Fluorescence images show the induction of LC3 puncta. Scale bar = 10 μm. (C, D) The calculated numbers of autolysosomes (red) and autophagosomes (yellow). Data are presented as the mean ± SD; $^{\#\#}P < 0.01$ and $^{\#\#\#}P < 0.001$ indicate between-group differences in autolysosomes (red); $^{*}P < 0.05$ and $^{**}P < 0.01$ indicate between-group differences in autophagosomes (yellow). One-way ANOVA followed by Bonferroni's test was used for statistical analysis. Representative images are displayed.

TLR2 knockdown. As shown in Fig 11B–11E, transfection of THP-1 macrophages with TLR2-siRNA significantly attenuated the levels of TNF-α, IL-6, IFN-γ and IL-1β.

## 4. Discussion

Recent studies have provided key insights into autophagy as being the fundamental mechanism to defend against invading pathogens in eukaryotic cells, with participation in pathogen clearance and immune defense. Autophagy appears to function as a double-edged sword for infections. On one hand, the evolution of many pathogens, such as *Salmonella*, varicella-zoster virus and *Listeria*, has involved various mechanisms for escape from autophagic degradation

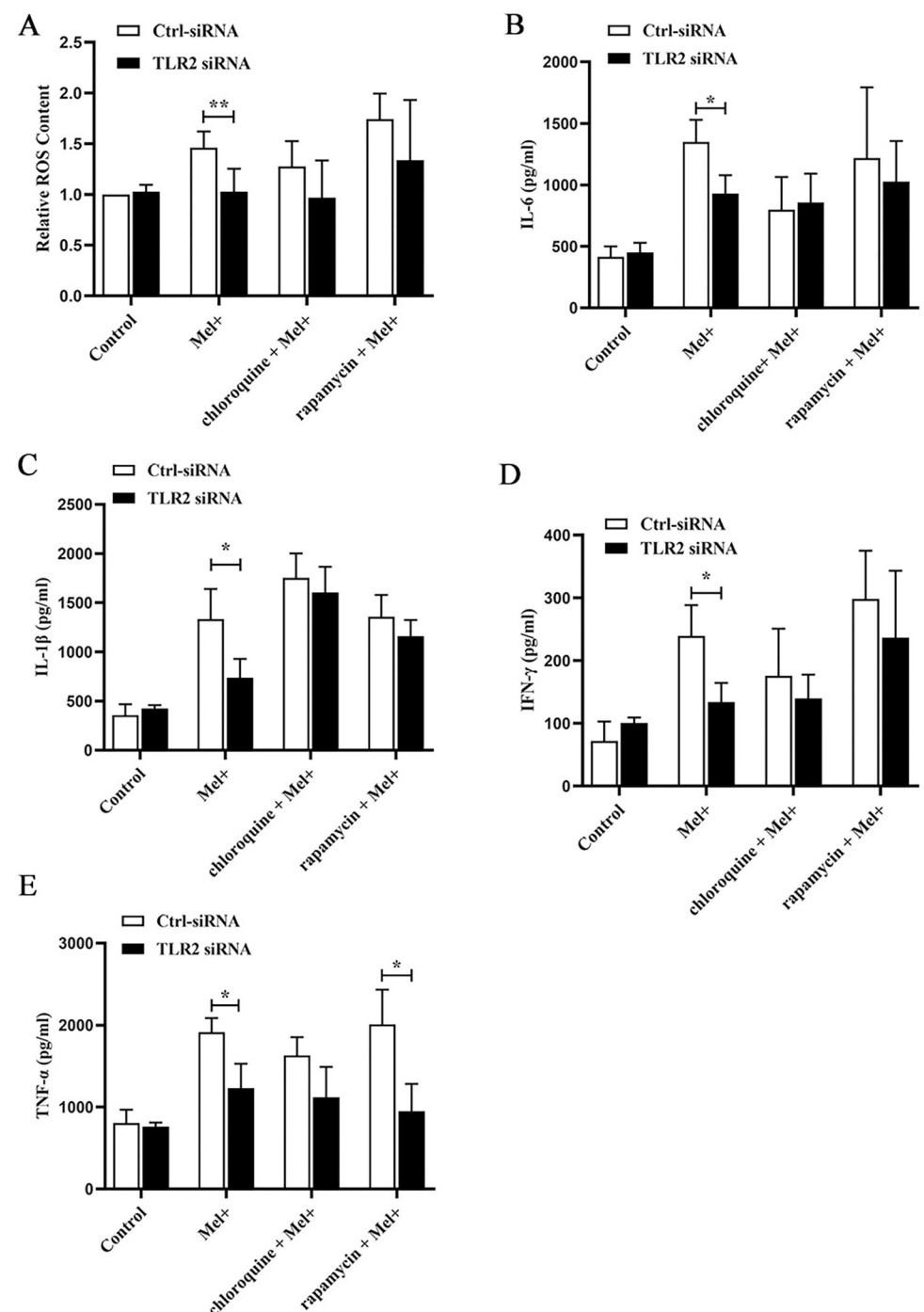

**Fig 11. The production of ROS and proinflammatory cytokines by macrophages transfected with TLR2 siRNA following exposure to *S. globosa*.** (A). ROS production in THP-1 macrophages transfected with TLR2-siRNA and pretreated with or without chloroquine (20 μM, 4 h) or rapamycin (5 μM, 12 h) before infection with Mel+ *S. globosa* conidia (MOI = 10) for 12 h. Data are shown as percentages relative to control cells, and as single datapoints (mean ± SD; n = 3); **$P < 0.01$ compared with the Ctrl-siRNA group. (B–E) Concentrations of inflammatory cytokines IL-6, IL-1β, IFN-γ and TNF-α in the THP-1 macrophage groups described in A. Data pooled from three sets of experiments are expressed as the mean ± SD; *$P < 0.05$, for the unpaired t-test.

and survival in the host cytoplasm [34–36]. On the other hand, autophagy also represents an underappreciated innate or adaptive defense mechanism for the control of intracellular pathogens that is involved in the pathogenesis of several diseases, including fungal infections [37]. Specifically, different species of fungal pathogens may behave differently under autophagy, but there are also different autophagy mechanisms within the cellular and host defense systems. Nicola and colleagues observed that LC3, a specific autophagosome marker, was present in most macrophage vacuoles containing *C. albicans*, whereas it was only present in a few vacuoles containing *C. neoformans*. Moreover, they also revealed that a lack of ATG5, a key protein in autophagic cell death, decreased the antifungal activity of macrophages against *C. neoformans* [14]. However, there are few reports on the role of autophagy during infection with non-opportunistic pathogenic fungi.

In the present study, the increased autophagic flux and protein levels of Beclin1 and LC3-II, together with the decreased levels of SQSTM1/p62 (sequestosome 1) proteins, suggested that *S. globosa* induced autophagy in THP-1 macrophages. Meanwhile, the killing assay revealed statistically significant interactions between the chloroquine and rapamycin groups regardless of the presence of melanin, which may indicate that autophagy can promote the clearance of *S. globosa* in THP-1 macrophages. Interestingly, these results also provide evidence that autophagy participates in host-defense activities against fungal pathogens, consistent with the finding of autophagy-mediated clearance in *A. fumigatus* reported by Kyrmizi and coworkers [38], suggesting that the killing of *A. fumigatus* was attenuated by silencing of the key autophagy protein ATG5. We also observed that treatment with autophagy activator rapamycin (which inhibits mTOR) [39, 40], chloroquine (which inhibits the acidification of autophagosomes and maturation of autolysosomes) [41] or wortmannin (which inhibits Ptdlns 3-kinase) [33] had an impact on autophagy-related proteins. As expected, rapamycin promoted autophagy flux, and this upregulation of autophagy was accompanied by the degradation of p62 and LC3-II. Meanwhile, both of the inhibitors further suppressed the *S. globosa*-mediated induction of autophagy, as evidenced by the higher levels of p62 in infected cells compared with those in control cells. In turn, identification of *S. globosa* is a key step in the occurrence of macrophage autophagy, and sequestosome-like receptors such as SQSTM1/p62 may capture pathogens into autophagosomes by combining with LC3 and *S. globosa* [42]. Autophagy has been shown to play multiple roles in the immune system, and is strongly associated with inflammation. Autophagy promotes cellular homeostasis by orchestrating inflammatory and immune responses, and at the same time, it also directly affects the production of proinflammatory cytokines from immune cells, acting at the levels of transcription, processing and secretion [43, 44]. In the present study, our results indicated that production of proinflammatory cytokines TNF-α, IL-6, IL-1β and IFN-γ was remarkably increased in *S. globosa*-infected macrophages. Interestingly, these proinflammatory cytokines play critical roles in antifungal immunity, with multiple roles ranging from autophagy to regulation of the inflammatory response [45]. Recently, we found that the levels of IFN-γ were significantly increased in patients with sporotrichosis (infected by *S. globosa*) compared with those in healthy controls, and subsequently promoted the T cells to deliver activating signals to effector phagocytes, enhancing phagocytic function and clearance of the fungal infection [46, 47]. Furthermore, Verdan and colleagues revealed that *S. schenckii*-infected bone marrow dendritic cells in mice strongly induced IL-6, TNF-α and IL-1β, leading to greater stimulation of IFN-γ production *in vitro* [48]. It is known that TNF-α and IL-6, as critical inflammatory mediators, can resist fungal infection and promote the infiltration of inflammatory cells to remove pathogens, such as increasing the killing of phagocytosed *S. schenckii* [49]. Consistent with this, in our previous findings, we verified that expression of the proinflammatory cytokines TNF-α and IL-6 increased during the phagocytosis of *S. globosa* by macrophages [28]. The present study found

that TNF-α, IL-6, IL-1β and IFN-γ were all induced in THP-1 macrophages infected with *S. globosa* conidia, and that three of these proinflammatory cytokines (TNF-α, IL-6 and IFN-γ) were positively related to autophagy in THP-1 macrophages. The levels of IL-1β were negatively associated with autophagic activity, indicating that the cytokines induced by *S. globosa* in THP-1 macrophages may be regulated by autophagy to produce a stronger host immune response. Since chloroquine could also increase the pH of phagolysosomes (which would presumably contain the conidia), the reduced levels of cytokine expression in chloroquine-treated cells may be related to impaired phagolysosomal processes, in addition to the inhibition of autophagolysosomal maturation. To confirm that the changes in cytokine levels were a direct consequence of autophagy, we used an siRNA against Atg7 to avoid interference by autophagic or phagocytic modulators. Atg7, as a core component of the autophagic machinery, is specifically involved in the activity of autophagy [50] and is a prerequisite for its induction [51], with knockdown of Atg7 thereby suppressing autophagic processes. Our results showed a decrease in proinflammatory cytokine production (TNF-α, IL-6 and IFN-γ) following the knockdown of Agt7. It was previously reported that inhibition of autophagy by knockdown of Atg16L1 in murine macrophages promoted the LPS-induced secretion of IL-1β through an inflammasome-dependent pathway, resulting in a decrease in bacterial clearance [52, 53]. Similarly, Crişan and coworkers demonstrated that inhibition of basal autophagy by 3-MA after stimulation with TLR2 or TLR4 ligands significantly promoted IL-1β, and decreased TNF-α production on human cells through downregulating transcription [54]. To a certain degree, these results are consistent with our findings that Atg7-knockdown THP-1 macrophages and cells pretreated with the autophagy inhibitor chloroquine (inhibiting the acidification of autophagosomes and the maturation of autolysosomes) enhanced the production of IL-1β. One possible explanation is that inhibiting autophagy might prevent degradation of pro-IL-1β, thereby leaving more cytokines in the cytosol available for processing and secretion [55]. These results may therefore indicate that inhibition of basal autophagy induces IL-1β overproduction. Moreover, autophagy has also been shown to be regulated by a diverse group of proinflammatory cytokines, with proinflammatory cytokines IFN-γ, IL-1, TNF-α, IL-17 and IL-6 reportedly inducing autophagy, and IL-13, IL-4 and IL-10 reportedly inhibiting autophagy [44, 56]. In addition to proinflammatory cytokines, our results demonstrated that *S. globosa*-induced autophagy increased ROS production in macrophages. ROS, as a crucial indicator of oxidative stress, are a major mechanism of macrophage killing of pathogenic microorganisms and signaling [57]. ROS accumulation leads to oxidative damage, which causes damage to DNA and disruption of the cell structure and function [58]. Compared with the control group, ROS generation was significantly higher in THP-1 macrophages infected with *S. globosa* conidia, consistent with the findings in *C. albicans* [59]. Furthermore, the production of ROS varied in the presence of autophagy modulators rapamycin and chloroquine, suggested that ROS participate in the process of autophagy and may be involved in the killing of *S. globosa*.

Melanin pigment, which exists as negatively charged multifunctional polymers, is an important virulence factor of phytopathogenic fungi [24]. Among its many functions, melanin confers protection against specific environmental stressors and prevents the production of antimicrobial oxidants during the host defense response [26, 60]. In this regard, melanin is also an antioxidant that protects against ROS. A variety of mechanisms have been proposed to explain the host–fungal melanin interactions [61, 62]. In our previous studies of the immune response to sporotrichosis, we demonstrated that *S. globosa* melanin could inhibit phagocytosis without the aid of an intermediary opsonin, and inhibit antigen presentation by downregulating major histocompatibility complex II expression in macrophages [27, 28]. However, prior to this study, we had not identified a role for autophagy in regulating the above immune processes. Herein, we extend our previous findings by demonstrating that melanin influences

the host immune response to macrophage autophagy, implying that distinct immune pathways might regulate host immune responses against fungal melanin. Our results showed that, not only was autophagy flux retarded in the presence of melanin, but that the protein expression of autophagy molecular markers (Beclin1 and LC3-II) was also inhibited, indicating that melanin had a negative effect on macrophage autophagy induced by *S. globosa*. Notably, another major human fungal pathogen, *A. fumigatus*, activates an LC3-associated non-canonical autophagy pathway during its interaction with macrophages [9]. Research in this area has shown that DHN-melanin synthesized by *A. fumigatus* can suppress the acidification and maturation of phagolysosomes by inhibiting the activity of the vacuolar-type ATPase (vATPase), thus interfering with the phagocytic and degradation process in macrophages [62]. However, vATPase activity is an indispensable component of the process of LC3 lipidation and distribution [63]. One plausible mechanism is that the type of DHN-melanin present in *S. globosa* has similar effects as that in *A. fumigatus* [26]. Upon further investigation of the effects of melanin on autophagy, our findings revealed that Mel+ conidia also inhibited the production of ROS and proinflammatory cytokines TNF-α, IL-6, IL-1β and IFN-γ, whereas Mel- conidia did not, in concordance with our previous studies on the effects of *S. globosa* melanin on phagocytic activity [28]. It is possible, therefore, that melanin inhibits free radical-mediated damage to impact the regulation of inflammation, thereby escaping the host immune response.

A number of recent studies have shown that TLRs are involved in pathogen defense via the induction of autophagy in macrophages [64]. Fang and colleagues [65] reported that siRNA knockdown of TLR2 remarkably reduced phagocytosis and autophagy during *Staphylococcus aureus* infection, showing that TLR2 provided an underlying mechanistic link between innate immune receptors and the induction of phagocytosis and autophagy in *S. aureus*-stimulated macrophages. Furthermore, several recent studies showed that TLR4 stimulation significantly enhanced the efficacy of *Mycobacterium tuberculosis* killing *in vitro* [66], and *M. tuberculosis*-infected macrophages activated with TLR4 agonists showed remarkably enhanced autophagy and bactericidal activity [67]. In the present study, even though *S. globosa* conidia significantly increased the expression of both TLR2 and TLR4, siRNA-mediated knockdown of TLR2, but not TLR4, was able to suppress autophagy in THP-1 cells, and this was dependent on substantially reduced protein expression levels of LC3-II and Beclin1, as well as autophagic flux. Additionally, because autophagy modulates the inflammation induced by TLR ligands in human cells [54, 55], knockdown of TLR2 expression significantly reduced the expression of proinflammatory cytokines TNF-α, IL-6 and IL-1β, whereas autophagy inhibitors failed to affect the production of these cytokines. It has been reported that TLR2 signals primarily induce inflammatory responses through the MyD88-dependent pathway [68], which may also be the signaling pathway by which autophagy regulators control inflammation, resulting in insignificant changes in TLR2 knockdown cells. Carlos and colleagues revealed that spleen cells in TLR2$^{-/-}$ mice were not able to release, or even decreased the production of, proinflammatory cytokines such as IFN-γ and IL-6 after infection with sporotrichosis [69]. Moreover, our findings indicated that autophagy regulators, such as chloroquine, only influence events downstream of autophagosomes and may therefore not be affected by TLR2, resulting in no clear trend in cytokine production. Our study further showed that inhibition of autophagy by knockdown of Agt7 led to the inhibition of inflammatory cytokine expression. However, the specific mechanisms involved require further elucidation. Analogous with this, Smeekens and colleagues revealed that blocking autophagy in the immune response against *C. albicans* in humans using pharmacological inhibitors did not affect important aspects of the *Candida*-induced immune response, such as cytokine production [18].

In conclusion, our study is the first to elucidate the importance of autophagy in the immune response of macrophages against *S. globosa*, as well as the role of melanin as an *S.*

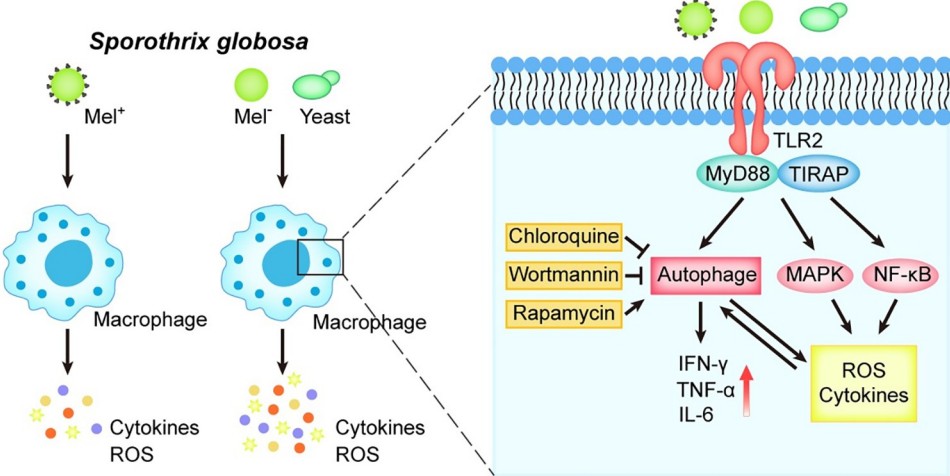

**Fig 12. *S. globosa* induces macrophage autophagy via TLR2.** TLR2 recognition of *S. globosa* triggers macrophage autophagy accompanied by ROS and proinflammatory cytokine production. Melanin inhibits autophagy by providing protection against oxygen-derived radicals, as well as suppressing the host proinflammatory cytokine response. TLR2 signaling also has the potential to induce ROS and proinflammatory cytokines, possibly by triggering NF-κB signaling or MAPK.

*globosa* virulence factor involved in resisting autophagy *in vitro*. This study also revealed that TLR2, as a cell surface receptor, may mediate *S. globosa*-induced autophagy in macrophages (Fig 12).

## Supporting information

**S1 Fig. Western blot analysis confirming Atg7 knockdown by siRNA.** (A) Western blotting of Atg7 was conducted in THP-1 macrophages with or without knockdown treatment. (B) The ratio of Atg7/GAPDH was calculated. $^*P < 0.05$, for the unpaired t-test. Representative images are displayed.
(TIF)

**S2 Fig. Increased expression of TLR2 and TLR4 in THP-1 macrophages upon *S. globosa* infection.** (A) THP-1 macrophages were treated with melanin ghosts (MGs), Mel+ or Mel- conidia (MOI = 10), Peptidoglycan (PGN) (20 μg/mL) and LPS (100 ng/mL) for 12 h. The protein levels of TLR2 and TLR4 were analyzed by western blotting. (B, C) The ratios of TLR2/GAPDH and TLR4/GAPDH were calculated. $^*P < 0.05$, $^{**}P < 0.01$, $^{***}P < 0.001$, for one-way ANOVA followed by Bonferroni's test. The results are representative of three experiments. Representative images are displayed.
(TIF)

**S3 Fig. Western blot analysis of THP-1 macrophages with TLR2 and TLR4 knockdown by siRNA.** Western blotting of (A) TLR2 and (C) TLR4 in THP-1 macrophages transfected with TLR2-siRNA, TLR4-siRNA or negative control-siRNA, then treated with Mel+ conidia (MOI = 10), PGN (20 μg/mL) and LPS (100 ng/mL) for 12 h. (B, D) The ratios of TLR2/GAPDH and TLR4/GAPDH were calculated. $^{***}P < 0.001$, for an unpaired t-test. Data are representative of three experiments and representative images are shown.
(TIF)

## Acknowledgments

We thank Michelle Kahmeyer-Gabbe, PhD, from Liwen Bianji (Edanz) (www.liwenbianji.cn) for editing the English text of a draft of this manuscript.

## Author Contributions

**Conceptualization:** Mengqi Guan, Lei Yao, Yan Cui, Shanshan Li.

**Data curation:** Mengqi Guan, Yang Song, Xiaobo Liu, Yan Cui.

**Formal analysis:** Mengqi Guan, Yu Zhen, Yan Cui.

**Funding acquisition:** Yan Cui, Shanshan Li.

**Methodology:** Mengqi Guan, Lei Yao, Yu Zhen, Yuanyuan Liu, Ruili Chen, Yan Cui, Shanshan Li.

**Supervision:** Yu Zhen, Yang Song.

**Writing – original draft:** Mengqi Guan, Lei Yao, Yan Cui, Shanshan Li.

**Writing – review & editing:** Mengqi Guan, Yan Cui, Shanshan Li.

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
