## [Decision Letter · Decision Letter 0]

27 Sep 2022

Dear Dr. Cui,

Thank you very much for submitting your manuscript "Sporothrix globosa melanin regulates autophagy via the TLR2 signaling pathway in THP-1 macrophages" for consideration at PLOS Neglected Tropical Diseases. As with all papers reviewed by the journal, your manuscript was reviewed by members of the editorial board and by several independent reviewers. In light of the reviews (below this email), we would like to invite the resubmission of a significantly-revised version that takes into account the reviewers' comments. 

The authors are commended for their thoughtful approach to this challenging pathogen. The manuscript has been reviewed by two experts in the field who have provided very thoughtful feedback on the work that require addressing point by point. Reviewer 2 especially highlights a concern about the duration of interaction, 24 hours, given the replication rate of the fungus. The authors should also consider more deeply addressing non-autophagy associated phagolysosomes and other immune signaling, especially in the context of the effects of rapamycin, chloroquine, and wortmannin.

We cannot make any decision about publication until we have seen the revised manuscript and your response to the reviewers' comments. Your revised manuscript is also likely to be sent to reviewers for further evaluation.

Sincerely,

Joshua Nosanchuk, MD

Section Editor

Joshua Nosanchuk

Section Editor

The authors are commended for their thoughtful approach to this challenging pathogen. The manuscript has been reviewed by two experts in the field who have provided very thoughtful feedback on the work that require addressing point by point. Reviewer 2 especially highlights a concern about the duration of interaction, 24 hours, given the replication rate of the fungus. The authors should also consider more deeply addressing non-autophagy associated phagolysosomes and other immune signaling, especially in the context of the effects of rapamycin, chloroquine, and wortmannin.

Reviewer's Responses to Questions

**Key Review Criteria Required for Acceptance?**

**Methods**

-Are the objectives of the study clearly articulated with a clear testable hypothesis stated?

-Is the study design appropriate to address the stated objectives?

-Is the population clearly described and appropriate for the hypothesis being tested?

-Is the sample size sufficient to ensure adequate power to address the hypothesis being tested?

-Were correct statistical analysis used to support conclusions?

-Are there concerns about ethical or regulatory requirements being met?

Reviewer #1: The objectives of the study seemed clearly articulated and the hypotheses were stated in the introduction. The methods and statistical analyses used in this manuscript seem appropriate in testing the hypotheses. However, I think clarifying the non-autophagy specific effects of chloroquine, rapamycin, and wortmannin and the limitations of the experiments would be helpful. 

Specific comments and questions regarding the methods section are:

Line 63: does “identified,” refer to identification as the S. globosa species or as a melanin mutant? I assume species. It would also be good to describe how the melanin-deficient mutant was either found or made in the lab (i.e. random mutagenesis?).

Line 72: Not sure if stimulated is the right word as two are inhibitors. 

Section 2.4 should include information about concentration of antibody used and how the western blots were quantified. 

Section 2.5 could benefit from explaining what the cutoff/threshold for the puncta size was (i.e. if small puncta were included, if so, how small?). A brief explanation, like the one in lines 145-146 about how this method works would be helpful. Also, it would be good to include exposure information. 

Line 102: Briefly state the premise of the assay, such as, “A dichlorofluorescein assay was used to measure ROS, where ROS in the cell oxidize the dye and cause it to fluoresce.”

Section 2.7: Provide information on how the cytokines were collected. From the supernatant I assume?

Reviewer #2: The study addresses an interesting subject worthy of research and publication. Most of the experimental design is sound and conclusions are supported by the results. However, I have the following concerns that need to be addressed to exclude alternative explanations and bias:

-The major flaw of the experimental design is the prolonged interaction times of up to 24 hours. It is unlikely that after this time conidia have not germinated and started to produce hypha (our calculations of conidia germination for this species are about 13-15 h), so the results at late times may be due to conidia germination instead of the suggested kinetic response. In the same line, what are the germination time and rate for both fungal strains, it is assumed it is the same, but this might be wrong. The authors have to demonstrate that both strains show similar germination rates. To minimize the potential effect of hypha production, the authors should perform the experiments with UV-killed cells; in this case, the cell wall is intact and viability lost. A good extra control that the authors should include in the experimental design is yeast-like cells, which do not synthesize melanin in the used experimental conditions. This will reinforce the role of melanin in this host-fungus interaction. There are different compounds that the authors added to the host-fungus interactions and it is not considered the possibility that this may contain bacterial LPS. The authors should demonstrate that these compounds are LPS-free. 

It is not clear why the authors only focused on TLR2 and TLR4, this is such a narrow repertoire of PRR involved in the interaction with the host and in mediating autophagy.

I do not think the parametric analyses used in the statistic section are the most appropriate ones for this study, non-parametric analyses should be included instead.

Finally, the English usage is acceptable but has room for improvement.

**Editorial and Data Presentation Modifications?**

Reviewer #1: It will be helpful to provide a summary diagram at the end of the manuscript to summarize and demonstrate the major findings of the paper and how they all come together. For example, you can show TLR2 signaling to autophagosome maturation to inflammatory cytokine and ROS release and where melanin interferes. 

For the bar graphs, it would be beneficial to include the individual data points in the graph, either as a scatter plot alone or overlayed on the bars themselves. The font on these graphs and on the microscopy graphs (Figure 2 and 7) are also small and might be difficult to read without zooming on the image (particularly in Figure 1), so increasing the text size would be good. 

The figure legends should include the information regarding which statistical tests were performed, as described in Methods section 2.9. For western blot images and microscopy images, it should be noted that the image is representative. The legend for the microscopy puncta counts should state the number of cells counted, as described in the methods section 2.5. 

There are a few instances in which the full figure is only referenced. It would help to reference the specific panels when the data is being discussed to help readers follow better, for example, Section 3.5. Additionally, sometimes the Figure/panels are only referenced once in the paragraph or section, while being discussed throughout, which can be difficult to follow.

Reviewer #2: (No Response)

PLOS authors have the option to publish the peer review history of their article (what does this mean?). If published, this will include your full peer review and any attached files.

Reviewer #1: Yes: Daniel F. Q. Smith

Reviewer #2: Yes: Héctor M. Mora-Montes

**Results**

-Does the analysis presented match the analysis plan?

-Are the results clearly and completely presented?

-Are the figures (Tables, Images) of sufficient quality for clarity?

Reviewer #1: Generally, the results are clearly presented and the figures have sufficient clarity. I think it may be hard to follow the story that is being told at times, so I think including a summary figure would be beneficial in ensuring the readers are on the same page by the end of the results section. 

Specific comments, questions, and edits regarding the results section are:

Lines 137, 141, 180: Add a sentence indicating what this pattern means. 

Section 3.1 does not reference Figure 1E,F. 

Figure 3: The western blot does not have a panel attributed to it. It might also help readers to reorder the quantifications to align with the order on the western blot. 

Section 3.3/Figure 4: Some of the changes described in this section are more difficult to see than the previous western blot figures. I think quantifying the western blot bands, as in Figure 1, would help strengthen the arguments made in 3.3 by making the differences and patterns clear. 

Line 205: I am not sure if mechanisms is the right word.

Line 209: Can remove, “the enhancement of“

Line 212: I don’t think “expression level” is the right word when measuring the direct quantity of ROS and not expression levels of a gene that produces the ROS. 

Line 241: Should only reference Supplementary Figure 1? 

Line 244: Should reference Supplementary Figure 2?

Figure 6: There are no statistics for Panel D. 

Figure 7: It might be better to break the figure into Panels A-D rather than just A and B. 

Lines 280-282: Awkward phrasing.

**Conclusions**

-Are the conclusions supported by the data presented?

-Are the limitations of analysis clearly described?

-Do the authors discuss how these data can be helpful to advance our understanding of the topic under study?

-Is public health relevance addressed?

Reviewer #1: The general premises of the manuscript that 1. S. globosa conidia induces autophagy and immune response mediated by TLR2, and 2. melanized conidia induce less autophagy and induce less cytokine/immune response, are supported by the data presented. I think the limitations (discussion of the non-autophagy related pathways that can activate the immune response, and discussion of the non-autophagy immune pathways that the "autophagy activators or inhibitors" can interfere with) are not discussed as extensively as they can, especially when interpreting the data. 

The public health and medical relevance of these findings are discussed in the introduction and discussion, as is the importance to the broader microbiology field studying host-fungal interactions. 

Specific questions and comments about the conclusions drawn about the data:

I think the results in 3.6/Figure 8 can be better explained, particularly explaining the significance of the trends in cytokine levels following TLR2 KD and treatment with chloroquine/rapamycin. 

Lines 287-289: I don’t think this statement is entirely correct. It seems that the treatment of chloroquine in the TLR2 KD infected cells did not decrease the levels of proinflammatory cytokines compared to the untreated TLR2 KD infected cells, nor does it seem that rapamycin increase proinflammatory cytokine production in the TLR2 KD infected cells, with the exception of IFN-gamma. Could you provide statistical evidence of this on the graph? Are the control (not TLR2 KD) different from the data in Figure 5? 

Lines 308-311 can be better explained. 

Line 318: I am not sure that you can state that it was protective, as survival outcome of the fungus was not looked at. 

Line 319: degradation is repeated. 

Line 320: Should it be “induction” instead of “suppression”?

Line 365: Besides reducing the production of ROS melanin is also an antioxidant and shields against ROS that is produced.

**Summary and General Comments**

Reviewer #1: Summary: 

The manuscript, “Sporothrix globosa melanin regulates autophagy via the TLR2 signaling pathway in THP-1 macrophages,” describes the induction of autophagy in macrophages, as measured by western blots and microscopy, following infection with S. globosa. These findings are interesting and further the understanding of how autophagy is involved in the immune response to fungal infections. Interestingly, the authors find that the induction of autophagy and subsequent effects are increased when the macrophages are infected with a non-melanin producing mutant. These findings add to the growing understanding of the host processes in which fungal melanins interferes, including interaction with TLR2. Overall, I think the authors could help provide clarity to their results with a schematic summarizing their findings (i.e. TLR signaling, autophagy, induction of inflammatory cytokines), and by describing how their findings can be differentiated from induction of inflammatory responses via non-autophagy related phagocytosis or non-autophagy related TLR signaling. 

Questions: 

Do you think the melanin-mediated reduction in autophagy is only related to less activation of TLR2 extracellularly or are there also effects downstream once the conidia are within the macrophage? 

Since chloroquine also increases the pH of phagolysosomes (which would presumably contain the conidia), do you think that the reduced levels of cytokine expression in chloroquine-treated cells can be related to impaired phagolysosomal processes in addition to inhibition of autophagolysosomal maturation? Is there a way to look at autophagy-specific downstream effects without also interfering with phagolysosomes (i.e. autophagy specific knockdowns)? I can see that autophagy is induced, and that can lead to immune activation. However, if the “autophagy inhibitors/activators” also interfere with non-autophagy immune signaling in the macrophages, then the degree to which autophagy is specifically responsible for this immune activation is unclear. 

Do you think ROS/cytokine production can be triggered by additional immune mechanisms outside of autophagy (i.e. NF-kB signaling, MAP Kinase)? This can be better described or illustrated in a summary figure. 

Does the induction of autophagy kill S. globosa? Can a survival experiment be done to show whether S. globosa survives better in autophagy deficient cells? Do melanized conidia get phagocytosed less than the non-melanized mutant? 

Do the conidia associate with the autophagolysosome in the macrophages?
---

## [Decision Letter · Decision Letter 1]

20 Feb 2023

Dear Dr. Cui,

Thank you very much for submitting your manuscript "Sporothrix globosa melanin regulates autophagy via the TLR2 signaling pathway in THP-1 macrophages" for consideration at PLOS Neglected Tropical Diseases. As with all papers reviewed by the journal, your manuscript was reviewed by members of the editorial board and by several independent reviewers. The reviewers appreciated the attention to an important topic. Based on the reviews, we are likely to accept this manuscript for publication, providing that you modify the manuscript according to the review recommendations. In particular, you are asked to address for one reviewer the potential issue for LPS contamination in your system and, for the second reviewer, challenges with the figures as detailed in their reviews. 

Sincerely,

Joshua Nosanchuk, MD

Section Editor

Joshua Nosanchuk

Section Editor

Reviewer's Responses to Questions

**Key Review Criteria Required for Acceptance?**

**Methods**

-Are the objectives of the study clearly articulated with a clear testable hypothesis stated?

-Is the study design appropriate to address the stated objectives?

-Is the population clearly described and appropriate for the hypothesis being tested?

-Is the sample size sufficient to ensure adequate power to address the hypothesis being tested?

-Were correct statistical analysis used to support conclusions?

-Are there concerns about ethical or regulatory requirements being met?

Reviewer #1: (No Response)

Reviewer #2: The authors addressed most of my concerns, and I am satisfied with the replies and new data, which make the whole dataset stronger. However, I still have a concern related to the potential contamination with LPS (Q2. There are different compounds that the authors added to the host-fungus interactions and it is not considered the possibility that this may contain bacterial LPS. The authors should demonstrate that these compounds are LPS-free). 

I have no doubts about the authors' aseptic technique. I am sure they are at the highest standards. However, the compounds used to interact with immune cells are often produced by genetics engineering using bacteria as a host, and the purification procedures may not get rid of LPS. Thus, I still think the authors should address this potential handicap in the experimental design.

**Results**

-Does the analysis presented match the analysis plan?

-Are the results clearly and completely presented?

-Are the figures (Tables, Images) of sufficient quality for clarity?

Reviewer #1: (No Response)

Reviewer #2: Because of the point already commented, results could be bias.

**Conclusions**

-Are the conclusions supported by the data presented?

-Are the limitations of analysis clearly described?

-Do the authors discuss how these data can be helpful to advance our understanding of the topic under study?

-Is public health relevance addressed?

Reviewer #1: (No Response)

Reviewer #2: Because of the point already commented, results could be bias.

**Editorial and Data Presentation Modifications?**

Reviewer #1: (No Response)

Reviewer #2: Nothing to add.

**Summary and General Comments**

Reviewer #1: I appreciate the authors’ responses to previous concerns. In particular, the addition of Figure 3 and Figure 7, both of which I find particularly interesting and supportive of the hypothesis/proposed model. On that note, I also think the summary figure is well-illustrated and helpful in summarizing the large amount of data presented. 

I still have some issues with the data presentation, as follows:

Figure 1 should probably be two separate figures. 

I find the legend for Figure 1 particularly unclear, and the legend makes it sounds like it is the same experiment in both halves even though it is not. 

 I think having better/larger labelling of Figure 1 will help with understanding. (I.e. if “Beclin”,”LC3-II”, etc were written above the relevant panels it will help in understanding what each column is quantifying). It might help to also label each row (I.e. write “Mel+” next to A and B). Also, it is important to label the X-axis to understand the graph better. 

The numbers and labelling of figures can generally be larger, but especially for Figure 1

Please reference specific panels in the text. For example, one reference to “Figure 5” should be split up into references to the individual panels in the discussion of the findings, when mentioned, throughout the rest of the paragraph. 

Similarly, Figure 7 should be split into references to individual panels throughout the paragraph rather than just referencing the entire figure in the final sentence of the paragraph. Currently that paragraph mostly seems in reference to the supplementary material rather than actual data presented in Figure 7 which only seems mentioned as an afterthought. 

Reference to 9B,D comes before 9A,C 

The GADPH Western Blot bands in Supplementary Figure 2A look more pixelated, geometrical, and darker than the rest of the bands presented in the manuscript. Were these the result of overexposure or a processing difference? If the bands were manipulated in a way, please make sure to mention that in the manuscript. 

Quantification data and unprocessed western blot image files do not seem to be available in the supplementary material.

Reviewer #2: Nothing to add.

PLOS authors have the option to publish the peer review history of their article (what does this mean?). If published, this will include your full peer review and any attached files.

Reviewer #1: No

Reviewer #2: No

Figure Files:

Data Requirements:

Reproducibility:

References

---

## [Decision Letter · Decision Letter 2]

28 Mar 2023

Dear Dr. Cui,

Thank you for the thoughtful manner in which you addressed the comments in the prior reviews. We are pleased to inform you that your manuscript 'Sporothrix globosa melanin regulates autophagy via the TLR2 signaling pathway in THP-1 macrophages' has been provisionally accepted for publication in PLOS Neglected Tropical Diseases.

Best regards,

Joshua Nosanchuk, MD

Section Editor

Joshua Nosanchuk

Section Editor

Reviewer's Responses to Questions

**Key Review Criteria Required for Acceptance?**

**Methods**

-Are the objectives of the study clearly articulated with a clear testable hypothesis stated?

-Is the study design appropriate to address the stated objectives?

-Is the population clearly described and appropriate for the hypothesis being tested?

-Is the sample size sufficient to ensure adequate power to address the hypothesis being tested?

-Were correct statistical analysis used to support conclusions?

-Are there concerns about ethical or regulatory requirements being met?

Reviewer #2: Thanks for addressing my concerns.

**Results**

-Does the analysis presented match the analysis plan?

-Are the results clearly and completely presented?

-Are the figures (Tables, Images) of sufficient quality for clarity?

Reviewer #2: Thanks for addressing my concerns.

**Conclusions**

-Are the conclusions supported by the data presented?

-Are the limitations of analysis clearly described?

-Do the authors discuss how these data can be helpful to advance our understanding of the topic under study?

-Is public health relevance addressed?

Reviewer #2: Thanks for addressing my concerns.

**Editorial and Data Presentation Modifications?**

Reviewer #2: Thanks for addressing my concerns.

**Summary and General Comments**

Reviewer #2: Thanks for addressing my concerns.

PLOS authors have the option to publish the peer review history of their article (what does this mean?). If published, this will include your full peer review and any attached files.

Reviewer #2: **Yes: **Héctor M. Mora-Montes

---

## [Editor Report · Acceptance letter]

29 Apr 2023

Dear Dr. Cui,

We are delighted to inform you that your manuscript, "*Sporothrix globosa* melanin regulates autophagy via the TLR2 signaling pathway in THP-1 macrophages," has been formally accepted for publication in PLOS Neglected Tropical Diseases.

Best regards,

Shaden Kamhawi

co-Editor-in-Chief

Paul Brindley

co-Editor-in-Chief
